# First Report of Tumor Treating Fields (TTFields) Therapy for Glioblastoma in Comorbidity with Multiple Sclerosis

**DOI:** 10.3390/brainsci12040499

**Published:** 2022-04-13

**Authors:** Rebecca Kassubek, René Mathieu

**Affiliations:** 1Department of Neurology, University of Ulm, 89081 Ulm, Germany; 2Department of Neurosurgery, German Federal Armed Forces Hospital Ulm, 89081 Ulm, Germany; rene.mathieu@uni-ulm.de

**Keywords:** case report, glioblastoma, multiple sclerosis, Tumor Treating Fields (TTFields), Optune^®^

## Abstract

Tumor Treating Fields (TTFields) therapy is FDA approved and has the CE mark for treatment of newly diagnosed and recurrent glioblastoma. To our knowledge, to date TTFields therapy remains unstudied in glioblastoma patients with multiple sclerosis (MS) as a comorbidity. Here, we present a patient who was diagnosed with MS at the age of 34. Treatment included several corticoid pulse treatments and therapies with interferon beta-1a and sphingosine-1-phosphate receptor modulator fingolimod. At the age of 52 the patient was diagnosed with glioblastoma, after experiencing worsening headaches which could not be attributed to the MS condition. After subtotal resection and concomitant radiochemotherapy, the patient received temozolomide in combination with TTFields therapy. For two years, the tumor condition remained stable while the patient showed high adherence to TTFields therapy with low-grade skin reactions being the only therapy-related adverse events. After two years, the tumor recurred. The patient underwent re-resection and radiotherapy and restarted TTFields therapy together with chemotherapy and is currently still on this therapy regime. Although having not been studied systematically, the case presented here demonstrates that TTFields therapy may be considered for newly diagnosed and recurrent glioblastoma patients with previously diagnosed multiple sclerosis.

## 1. Introduction

Glioblastoma (GBM) is the most common malignant primary brain tumor in adults and shows high malignancy with a median survival of 8 months, regardless of treatment [1], assessed in a large population-based cancer registry in the United States of America. With the incidence of GBM increasing with age, the peak is at 75–84 years [1]. The established standard of care for patients with newly diagnosed GBM is maximal safe surgical resection followed by radiation therapy (RT) with concomitant temozolomide (TMZ) and subsequent adjuvant TMZ therapy +/− TTFields therapy [2]. However, survival rates remain low with a 7.2% 5-year survival rate [1].

TTFields are a non-invasive anti-mitotic treatment modality based on the continuous delivery of low intensity (1–3 V/cm), intermediate frequency (100–300 kHz), alternating electric fields to the region of a tumor [3]. TTFields interfere with mitosis by causing dipole alignment of tubulin and septin molecules with the electric field during metaphase and anaphase leading to hampered mitotic spindle and cleavage furrow formation [3,4,5]. Additionally, the hourglass cellular structure during telophase induces non-uniform fields within mitotic cells, creating an area of high field intensity at the cleavage furrow. In a process called dielectrophoresis, polar molecules are drawn towards the region of high field intensity resulting in uneven distribution of cellular components and chromosomes, which potentially induces cell death [3,4,5,6]. The application of TTFields using Optune^®^ during maintenance treatment with TMZ for newly diagnosed GBM was demonstrated to be effective and safe in a large phase III trial [7].

Multiple sclerosis (MS) is a potentially severe cause of neurological disability in young adults with a more than 10% increase in the age-standardized prevalence in recent decades. The inflammatory disease affects the central nervous system and presents heterogeneously depending on affected areas, similar to GBM. Treatments comprise acute relapse management and symptomatic treatments, and a rapidly growing number of disease-modifying medications are available to reduce the frequency of episodes of neurologic disability and limit the accumulation of focal lesions on magnetic resonance imaging [8,9].

Concurrent diagnosis of GBM and MS is uncommon and not systematically studied to date [10]. Several case reports of MS patients with brain tumors, including GBM, were published [11,12,13,14,15,16,17,18]. However, to our knowledge, there is no report of an MS patient with GBM being treated with TTFields yet.

Here, we report the first case of a patient with MS treated with TTFields for GBM showing high therapy adherence.

## 2. Case Report

The patient was diagnosed with MS at the age of 34 (Figure 1), showing optic neuritis, somatosensory deficits, and vertigo. Treatment included several corticoid pulse treatments and therapies with interferon beta-1a and sphingosine-1-phosphate receptor modulator fingolimod due to myelitis with progressive paraspasticity, loss of lower extremity proprioception, and loss of bladder and bowel control. Therapy was well tolerated with exception of leucopenia. However, due to progressive spinal lesions, disability worsened though therapy with fingolimod was continued until the diagnosis of malignant glioma at the age of 52.

The 52-year-old patient with relapsing-remitting multiple sclerosis (RR-MS) and a Karnofsky performance status of 70% presented with progressive gait disturbance and a declining general condition. Associated to the MS condition, a spastic paraparesis and left-sided limb ataxia was present. Moreover, the patient experienced worsening headaches leading to vomiting in one event, which at this point was attributed to an MS relapse and treated with intrathecal administration of glucocorticoids. Since the patient’s condition did not improve upon treatment, an MS unrelated intracranial space-occupying lesion was considered. Magnetic resonance imaging (MRI) revealed a space-occupying lesion in the right hemisphere with contrast-enhanced margin spreading from fronto-parietal to precentral cortex and septum pellucidum, approximately 4.8 × 4.4 × 3.8 cm^3^ (Figure 2).

The patient underwent fluorescence-guided subtotal resection with 5-ALA according to Stummer et al. [19] using intraoperative magnetic resonance imaging (ioMRI) [20]. Gross total resection (>95% of tumor volume) is the aim in brain tumor surgery and could nearly be achieved in this patient: the residual tumor volume was 4.86 cm^3^, i.e., 6% of the initial volume of 80.26 cm^3^. Newer studies have shown that a “safe maximal tumor resection” compared to a strict gross total resection improves quality of life, overall survival, and progression free survival [21]. With the fluorescence guided technique and an intraoperative MRI, the tumor infiltrated area of the corpus callosum has been spared and a safe maximal tumor resection was achieved.

Histopathology and molecular diagnostics led to the diagnosis of GBM with absent IDH mutation or 1p19q-codeletion, 63.8% MGMT-promoter methylation, and ATRX mutation. The tumor was positive for GFAP expression as well as areas of MAP-2 expression. Ki-67 labeling revealed a proliferative index of up to 20%. While giant cells were not present, large necrotic areas were detected.

Subsequent to the resection, the patient received radiotherapy (60.0 Gy) with concomitant daily TMZ chemotherapy. Chemotherapy had to be stopped after 5 weeks due to haematotoxicity with thrombocytopenia and was restarted after a break of 4 weeks after completion of radiotherapy with a reduced dose of 150 mg/m² body surface area together with the initiation of TTFields therapy. Chemotherapy was terminated after 11 cycles, while TTFields therapy was continued for another 8 months as monotherapy. During approximately 2 years of stable tumor disease following primary GBM resection, the patient showed an average TTFields usage of over 80%, clearly beyond the suggested threshold of 75% (Figure 3). MS therapy with fingolimod was ceased at the time of GBM diagnosis; although no specific MS therapy was administered since then, no clinical bouts occurred and no new MS lesions could be detected in MRI. However, the general clinical status of the patient slightly deteriorated causing decreased functional status due to the pre-existing neurological deficits.

Approximately 2 years after primary GBM resection and stable tumor condition, a routine MRI scan showed an area of contrast-enhancing tissue and recurrent GBM was identified in the right frontal lobe (Figure 4). Hence, TTFields therapy was interrupted, and the patient underwent fluorescein-guided subtotal resection according to Acerbi et al. [22] followed by radiotherapy. After completion of radiotherapy, the patient restarted chemotherapy with TMZ, again with a reduced dosage, accompanied by TTFields therapy. After 5 cycles of TMZ, a new contrast-enhancing lesion in the corpus callosum was detected, highly suggestive for recurrent glioblastoma. Chemotherapy was switched to lomustine in combination with TTFields therapy at this time according to the patient’s informed consent to this individualized therapy. The patient is currently still on this combined therapy. Until today, no TTFields therapy-related adverse events except skin irritations and erosions were observed, managed by topical corticosteroids and not leading to treatment interruption.

## 3. Discussion

Treatment of GBM, the most common and highly malignant primary brain tumor in adults, remains challenging and treatment entities are limited. In the multi-center, randomized, open-label phase III EF-14 study, the efficacy and safety of continuous TTFields application during maintenance treatment with TMZ for newly diagnosed GBM was demonstrated [7]. Median overall survival and survival rates were significantly increased in the TTFields-treated patients compared to TMZ treatment alone, with a 5-year survival rate of 13% vs. 5%. There are no studies providing information on the interaction between TTFields therapy and MS, because TTFields therapy has only been studied in patients with different cancer entities. In general, there are no data available on possible interferences of electromagnetic fields on MS. Due to exclusion of patients with neurological disorders from the trial, TTFields treatment of GBM patients with comorbid MS has not been studied systematically until today. Here, we report the first case of a GBM patient with MS comorbidity being treated with TTFields. The patient has been on TTFields therapy for over 3 years showing high therapy adherence, only interrupted by a treatment break of approximately 4 months due to re-resection and re-irradiation. Subgroup analyses of the EF-14 phase III trial revealed that continuing TTFields treatment beyond first progression was associated with a significantly prolonged median overall survival compared to second-line chemotherapy alone [23]. Moreover, a threshold of 50% daily TTFields usage in combination with TMZ was associated with a significant improvement of median overall survival (OS) and progression-free survival (PFS) compared to TMZ alone [24]. Increased compliance is an independent prognostic factor for improved survival of GBM patients and does not depend on gender, age, MGMT-promoter methylation status, degree of resection, and performance status [24]. The high usage by the patient indicates successful integration of the TTFields therapy into the daily routine, also beyond tumor progression. Moreover, besides manageable low-grade skin reactions, no therapy-related adverse events were observed, confirming a good therapy safety profile [7,25]. Post hoc subgroup analyses of the EF-14 trial revealed an increase in overall survival also for patients with glioblastoma and methylated MGMT promotor. Our patient’s survival of more than three years is more than the median overall survival of patients with glioblastoma and methylated MGMT promotor with standard therapy, ranging from 21.7 up to more than 30 months [26,27,28], mostly due to a better response to alkylating chemotherapy. Conclusions about a potential association with the use of TTFields therapy cannot be drawn from this single case observation.

Although specific MS therapy was terminated at the time of GBM diagnosis, the patient did not show any additional clinical bouts and no new MS lesions were detected in MRI while being on TTFields therapy. However, the general clinical status of the patient slightly worsened causing decreased functional status.

## 4. Conclusions

In conclusion, our case shows that TTFields therapy may also be considered for newly diagnosed and recurrent GBM patients with co-morbid MS. Since TTFields treatment has not been studied in this small concurrently diseased population to date, further reports are needed to confirm our observation.

## Figures and Tables

**Figure 1 brainsci-12-00499-f001:**
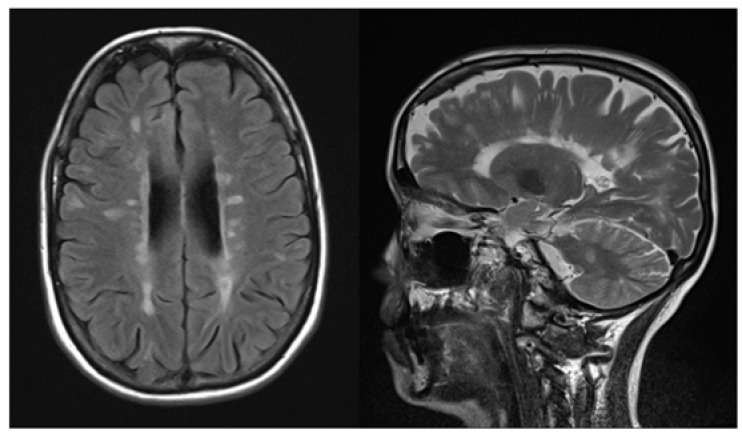
MRI scans showing high load of MS lesions one year prior to GBM diagnosis (**left**: axial FLAIR, **right**: sagittal T2w).

**Figure 2 brainsci-12-00499-f002:**
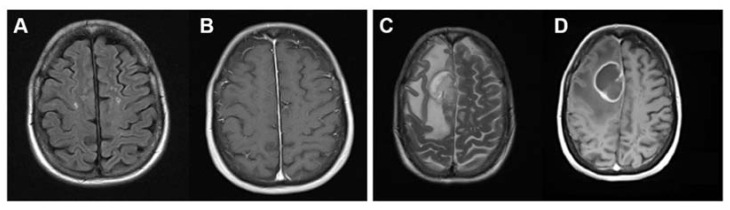
MRI scans taken prior to GBM diagnosis (**A**: FLAIR, **B**: ce T1w) and 9 months later at the time of GBM diagnosis (**C**: T2w, **D**: ce T1w).

**Figure 3 brainsci-12-00499-f003:**
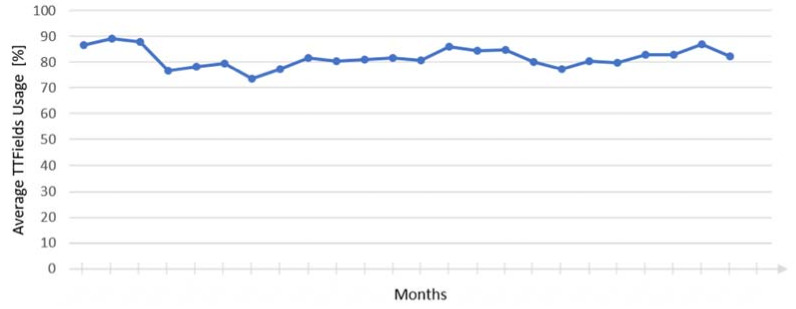
Average usage of TTFields therapy before tumor recurrence.

**Figure 4 brainsci-12-00499-f004:**
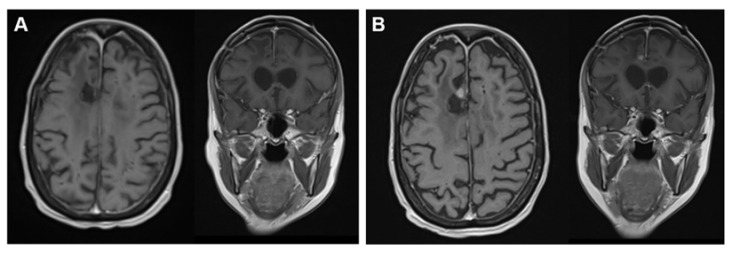
MRI scans showing stable condition approximately 2 years after diagnosis (**A**: ce T1w) and 3 months later at the time of tumor recurrence (**B**: ce T1w).

## Data Availability

Not applicable.

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
