# Peer review of "First Report of Tumor Treating Fields (TTFields) Therapy for Glioblastoma in Comorbidity with Multiple Sclerosis"

_brainsci, 2022, doi:10.3390/brainsci12040499_

Round 1

Reviewer 1 Report

This article reports a patient case Tumor Treating Fields (TTFields) therapy for glioblastoma in comorbidity with multiple sclerosis (MS). It is advisable to make the introduction shorter and to broaden the discussion. The authors should clarify why a subtotal resection was carried out and the percentage of tumor residue. It is also advisable to indicate the expected survival based on the patient's prognostic factors and if this resulted greater than expected. It is necessary for the authors to explain the interaction between TTFields therapy and MS.

Author Response

Please find in the following a point-by-point response to all comments. All changes to the text are highlighted in the submitted revised manuscript.

1) It is advisable to make the introduction shorter and to broaden the discussion.

Reply: Thank you for this comment. Accordingly, the introduction was shortened and the discussion was broadened.

2) The authors should clarify why subtotal resection was carried out and the percentage of tumor residue.

Reply: To clarify this item the following paragraph was added:

“… Gross total resection (>95 % of tumor volume) is the aim in brain tumor surgery and could nearly be achieved in this patient: the residual tumor volume was 4.86 cm³, i.e., 6% of the initial volume of 80.26 cm³. Newer studies have shown that a “safe maximal tumor resection” compared to a strict gross total resection improves quality of life, overall survival, and progression free survival [21]. With the fluorescence guided technique and an intraoperative MRI, the tumor infiltrated area of the corpus callosum has been spared and a safe maximal tumor resection was achieved.”

3) It is also advisable to indicate the expected survival based on the patient´s prognostic factors and if this resulted greater than expected.

Reply: To address this issue we added the following part to the discussion:

“Post hoc subgroup analyses of the EF-14 trial revealed an increase in overall survival also for patients with glioblastoma and methylated MGMT promotor. Our patient´s survival of more than three years is more than the median overall survival of patients with glioblastoma and methylated MGMT promotor with standard therapy, ranging from 21.7 up to more than 30 months [26,27,28], mostly due to a better response to alkylating chemotherapy. Conclusions about a potential association with the use of TTFields therapy cannot be drawn from this single case observation.” 

4) It is necessary for the authors to explain the interaction between TTFields therapy and MS.

Reply: There are no data available, as now reported in the Discussion, reading,

“There are no studies providing information on the interaction between TTFields therapy and MS, because TTFields therapy has only been studied in patients with different cancer entities. In general, there are no data available on possible interferences of electromagnetic fields on MS.”

Reviewer 2 Report

It is a case study, however, could you please add studies about possible interferences or beneficial actions of electromagnetic fields / pulsed fields on MS patients. 

Author Response

Please find in the following a point-by-point response to all comments. All changes to the text are highlighted in the submitted revised manuscript.

It is a case study, however, could you please add studies about possible interferences or beneficial actions of electromagnetic fields / pulsed fields on MS.

Reply: There are no data available, as now reported in the Discussion, reading,

“There are no studies providing information on the interaction between TTFields therapy and MS, because TTFields therapy has only been studied in patients with different cancer entities. In general, there are no data available on possible interferences of electromagnetic fields on MS.”